# Size-dependence in chord characteristics from simulated and observed continental shallow cumulus

Philipp J. Griewank[1], Thijs Heus[2], Neil P. Lareau[3], and Roel A. J. Neggers[1]

[1]University of Cologne, Germany
[2]Cleveland State University, Ohio, USA
[3]University of Nevada, Reno, USA

**Correspondence:** Philipp J. Griewank (philipp.griewank@uni-koeln.de)

**Abstract.**

In this study we compare long-term Doppler and Raman lidar observations against a full month of large eddy simulations of continental shallow cumulus clouds. The goal is to evaluate if the simulations can reproduce the mean observed vertical velocity and moisture structure of cumulus clouds and their associated subcloud circulations, and to establish if these properties depend on the size of the cloud. We propose methods to compare continuous chords of cloud detected from Doppler and Raman lidars with equivalent chords derived from 1D and 3D model output. While the individual chords are highly variable, composites of thousands of observed and millions of simulated chords contain a clear signal. We find that the simulations underestimate cloud size and fraction, but successfully reproduce the observed structure of vertical velocity and moisture perturbations. There is a clear scaling of vertical velocity and moisture anomalies below the chords with chord size, but the moisture anomalies are only 1-2 % higher than the horizontally mean values. The differences between the observations and simulations is smaller than the difference in sampling the modelled chords in time or space. The shape of the vertical velocity and moisture anomalies from cloud chords sampled spatially from 3D model snapshots is almost perfectly symmetric. In contrast, the chords sampled temporally from the lidar observations and 1D model output have a marked asymmetry with stronger updrafts and higher moisture anomalies occurring earlier on.

## 1 Introduction

Shallow cumulus cloud populations contain a wide range of spatial, temporal, and physical/dynamical variability. The individual clouds themselves vary greatly in size, life time, depth, and micro-physical properties. But even statistics of large shallow cumulus populations differ as well, with sub-mesoscale dynamics, such as cold pools, causing large-scale organization and aggregation (Sengupta et al., 1990; Nair et al., 1998). This high individual and statistical variability, combined with their comparatively small temporal and spatial scales, make shallow cumulus a tricky aspect of the atmosphere to represent numerically in weather and climate models (e.g. Neggers and Siebesma, 2013; Nam et al., 2014), from which they can not be neglected due to their critical contribution to Earth's energy and water budget (e.g. Sherwood et al., 2014; Bony et al., 2015; Brient and Schneider, 2016).

One useful paradigm that simplifies representing the effects of sub-grid scale shallow cumulus on resolved-scale flow is the assumption that various properties of shallow cumulus are size-dependent, which enables representing the high variability in cumulus clouds through a limited number of shallow convective plumes of differing size (e.g. Arakawa and Schubert, 1974; Neggers, 2015; Olson et al., 2019). The idea that larger shallow cumulus clouds have stronger updrafts than small shallow cumulus clouds is as intuitive as it is old (e.g. Plank, 1969; Raga et al., 1990; Benner and Curry, 1998; Zhao and Girolamo, 2007; Yuan, 2011). Assuming shallow cumulus clouds are created by buoyant plumes which are slowed via entrainment with the dry surrounding air (Turner, 1962; Simpson and Wiggert, 1969; Warner, 1970a), it follows that larger plumes could rise faster by either being more buoyant, entraining less, or a mix of both. The assumed size-dependence of plume entrainment is a corner stone of the first bin-macrophysics approach in spectral modeling introduced by Arakawa and Schubert (1974). Model studies support this entrainment dependence (e.g. Zhao and Austin, 2005; Peters et al., 2020), and the dependence is still being used today (e.g. Neggers, 2015; Olson et al., 2019).

The potential use of size-dependence in convective parameterization has motivated historic and recent efforts to observe the existence of this dependence in nature, but detecting this dependence in observations has proven to be quite difficult. This is in part because due to the relatively small size of shallow cumulus, which predominantly have a horizontal area below a single square km, the resolution of 3D scanning radars is insufficient to fully resolve the associated wind field. The short life span and non-stationary character further complicate measurements, as only a short time window exists to measure the active updraft. Airplanes can fly right through the center of cumulus clouds to measure a 1D track though the cloud, and have a long and successful history studying the structure of cumulus clouds and their subsiding shells, especially for maritime cumulus (e.g. Warner, 1970b; French et al., 1999; Laird et al., 2000; Rodts et al., 2003; Neggers et al., 2006; Heus and Jonker, 2008; Wang and Geerts, 2010, 2011). However, the high operating costs of aircraft means that for practical reasons the total number of clouds which can be sampled via aircraft is limited, and commonly only a 1D line is sampled along a specific height. Other novel approaches such as tethered balloons (Kitchen and Caughey, 2007), helicopters (Siebert et al., 2006), or radar chaff detected from a plane mounted radar (Jung and Albrecht, 2014) suffer from similar practical constraints. Given the large role turbulence and individual thermals play in shallow cumulus development, many samples are needed to detect if a size dependence is present.

This is where upward facing lidar observations excel. After years of steady deployment of Doppler lidars at the ARM-SGP (Atmospheric Radiation Measurement - Southern Great Plane) and other super-sites, thousands of shallow cumulus have been measured as they pass over lidars. While the Doppler lidars can not measure above the boundary layer or inside the cloud, they have sufficiently high temporal and spatial resolution to resolve the vertical velocity below shallow cumulus clouds. Both Lamer and Kollias (2015) and Lareau et al. (2018) successfully collected over a 1000 Doppler lidar observations of shallow cumulus clouds at the ARM-SGP site which showed that updraft strength increases with cloud chord length. In regards to the spatial structure, Lareau et al. (2018) show that on average a clear 2D updraft exists below the cloud, and Lareau (2020) achieved a similar result using Raman lidar measurements of water vapor.

In contrast to observations, Large Eddy Simulations (LES) of shallow cumulus in principle allow for unlimited sampling of all simulated properties in 4D. Studies such as Dawe and Austin (2012); Böing et al. (2012), and Neggers (2015) have taken

advantage of this fact to study the cloud properties of maritime cumulus in individual LES. However, LES still represent a simplification of reality and should be critically compared to measurements where possible. But bringing together LES and observations on cumulus statistics, in particular detailed dependencies on size, require both large amounts of observations and LES. This is especially true for continental shallow cumulus that have a marked daily cycle and strong day to day variability. But

thanks to ongoing efforts to run semi-continuous LES for observational super-sites in Europe (Neggers et al., 2012; Schalkwijk et al., 2015; van Laar et al., 2019) and the US (LASSO (LES ARM Symbiotic Simulation and Observation) Gustafson et al., 2018, 2020) we now have the ability to robustly compare observed and simulated cloud statistics, as done recently for cloud base vertical velocity by Endo et al. (2019). Maritime cumulus comparisons require fewer simulations, as shown by the analysis of cloud base mass flux at the Barbados Cloud Observatory of Sakradzija and Klingebiel (2020).

In this study we take advantage of the newly available large amounts of high-resolution simulations and long-term lidar observations. We do so by linking the observational methods of Lareau et al. (2018) and Lareau (2020) with the LASSO strategy of running LES of all shallow cumulus days from 2015 to 2018 at the ARM-SGP site (Gustafson et al., 2018, 2020). In order to get a resolution of 25 m, which is higher than the default LASSO settings, we use the LASSO forcing data to simulate 28 cumulus days with the MicroHH model (van Heerwaarden et al., 2017).

The main goals of this paper are (1) to determine the size-sensitivity of subcloud vertical velocity and moisture anomaly fields, and (2) to determine if the MicroHH simulations provide a reliable approximation of shallow cumulus cloud statistics. The goals are accomplished by making a one-to-one comparison of observed and simulated cloud chords and their subcloud circulation. To achieve this one-to-one comparison we sample chords from 1D column output, which capture the temporal evolution of clouds as they pass over a fixed point, and from 3D model output snapshots, which are frozen in time. After

accomplishing the main goals of this paper we use our large data set to quantify the sampling uncertainty of various observed chord properties as a function of lidar deployment days.

This paper begins with an explanation of the data and simulations, followed by a comparison of simulated and observed cloud fraction and cloud base height at the ARM-SGP site in section 2. We then detail the methods for extracting chords from 1D (3.1) and 3D (3.2) model output to compare against the lidar observations. Before we evaluate the scale dependence and

shape of vertical velocity and moisture anomalies in sections 5 and 6, we first compare the observed and simulated chords distributions in section 4 to test how representative the simulated chords are and how much our results depend on the precise chord definitions used. We end our results with an analysis on the sampling uncertainty in section 7 in which we quantify how the uncertainty decreases for various variables when more days of lidar observations are available. We end with a conclusion, and discussion in section 8.

## 2 Observations and simulations

### 2.1 Observations

In this subsection we provide a brief overview of the instrumentation used to observe the vertical velocity and water vapor mixing ratio. For a more in depth description of the data and how they were post-processed we refer the reader to Lareau (2020) (vertical velocity) and Lareau (2020) (water vapor mixing ratio).

#### 2.1.1 Vertical velocity

Subcloud and cloud-base vertical velocity are observed with a network of 5 Doppler lidars (DL) located at ARM-SGP Newsom (2010). The lidar network is arranged with 4 outlying sites (E32,E37,E39,E41) and one central location (C1). The outlying sites fall on an approximately 50 km radius circle centered on C1. The DLs are active ground-based infrared (1.5 μm) laser remote sensors that provide range- and time-resolved profiles of the line-of-sight velocity (i.e., the vertical velocity when pointed vertically) and the attenuated backscatter coefficient (hereafter backscatter) over a range of 9-12 km from the surface (depending on the lidar and its settings). The lidar data are processed at 30 m range gate resolution and 1.3 sec temporal resolution. The DL is sensitive to micron scale aerosol, which provide a tracer for boundary layer flows. As such the lidars provide an ideal tool for resolving the time-height structure of large convective eddies in the boundary layer. The lidar beams also rapidly attenuate in liquid water, thereby enabling cloud-base detection. The DL is also used to identify the time varying CBL height, based on a threshold of the vertical velocity variance computed for 15 minute intervals (Tucker et al., 2009). Here we use a variance threshold of 0.08 $m^2$ $s^{-2}$, which Lareau et al. (2018) show produces a good representation of CBL heights during ShCu conditions.

#### 2.1.2 Water vapor mixing ratio

Subcloud and cloud-base water vapor mixing ratio (q) is determined using an Ultraviolet (UV) Raman Lidar (RL), which is located adjacent to the DL at C1. The RL is sensitive to both molecular and aerosol backscatter, with the molecular backscatter used in retrieval of the water vapor mixing ratio (Wulfmeyer et al., 2010). The retrieved water vapor profiles are available at 10 second temporal- and 50 m spatial-resolution (RAMAN LIDAR Vertical Profiles 10SRLPROFMR1TURN). The ability of RL to measure the first through third moments of the boundary layer water vapor mixing ratio is well-established (Wulfmeyer et al., 2010; Turner et al., 2014a) and the RL at ARM-SGP has been validated against aircraft data (Turner et al., 2014b).

### 2.2 Model

We are basing our analysis on cloud fields generated with MicroHH (van Heerwaarden et al., 2017). This Large Eddy Simulation model has been validated against a wide range of standard cases, including shallow cumulus intercomparsion cases in marine (e.g. BOMEX, Siebesma et al., 2003) and continental (e.g. ARM, Brown et al., 2002) conditions. For the current study, we simulated 28 days with shallow cumulus convection over the Department of Energy's Atmospheric Radiation Measure-

**Table 1.** Dates of simulations included in the analysis (in Month/Day).

| 2015 | 06/06 | 06/09 | 06/27 | 08/01 | | |
|------|-------|-------|-------|-------|-------|-------|
| 2016 | 05/18 | 05/30 | 06/10 | 06/11 | 06/19 | 06/25 |
|      | 07/19 | 07/20 | 08/18 | 08/19 | 08/30 | |
| 2017 | 05/09 | 06/05 | 06/27 | 07/04 | 07/16 | 07/19 |
|      | 07/20 | 07/22 | | | | |
| 2018 | 05/22 | 06/06 | 07/05 | 07/09 | 07/10 | |

ment site in the Southern Great Plains (ARM-SGP), based on the Large-Eddy Simulation (LES) ARM Symbiotic Simulation and Observation (LASSO; Gustafson et al., 2017) database. These realistic and routine simulations of cumulus fields over the ARM-Southern Great Plains observatory in Oklahoma are run using a variety of initial conditions and model settings. For each day in the list in Table 1, we selected the configurations with the best match to the observations in cloud cover and liquid water

path. We ran one simulation for each day in the LASSO version 1 release, as long as the cloud cover and liquid water path have some skill in the best matching simulation, according to the LASSO Bundle Browser (https://adc.arm.gov/lassobrowser) (skill scores above 0.3). Since the simulations in the LASSO database were done on a relatively coarse resolution of 100 m, and with insufficient temporal output frequency to mimic lidar observations, we re-ran all cases with MicroHH on a 25 m resolution in all directions and a horizontal domain size of 25.6 x 25.6 km$^2$. Between a height of 6 km to the domain top at 9 km, the

vertical grid stretches from 25 m to 150 m. Adaptive time stepping with a constant Courant-Friedrichs-Lewy criterion results in a time step typically between 1 and 2 seconds. The simulations were run with periodic boundary conditions, homogeneous and prescribed surface fluxes, and a prescribed radiative tendency profile. Two moment warm microphysics was used, even though precipitation was negligible. Average statistics and 3D output of all thermodynamic variables were provided every 1800 s. Raw column data was provided every time step (i.e., about every second) at $4^2 = 16$ locations throughout the domain, with

each column spaced 6.4 km apart in both the x and y direction.

## 2.3 Simulation evaluation

Before beginning our analysis of cloud chords we first determine to what degree our simulated cloud-topped boundary layer matches the observations by comparing cloud base height and cloud fraction. Thanks to the evaluation data provided as part of the LASSO library along with the forcing data, we can easily compare the hourly cloud base height and cloud fraction.

We find that the cloud base height of simulations and observations agree well, with the average simulated cloud base being only 35 m lower than the observed (Figure 1). As expected, the boundary layer deepening over the course of the day is clearly visible in the individual simulations. This good agreement of cloud base height matches was also found for the original LASSO simulations (Gustafson et al., 2020), and can be directly attributed to the forcing data provided by the LASSO project being well calibrated.

In contrast to cloud base height, the modelled and observed cloud fraction align less well (Figure 2). The observed hourly cloud fraction has a higher temporal variability, with measurement-to-measurement changes of up to 0.5 (e.g. 20170716 Figure 2). Such strong shifts likely represent the sampling bias in a spatially and temporally heterogeneous cumulus topped boundary layer (Rossow, 1989). Since the cloud fraction from the MicroHH simulations is calculated from 3D snapshots of the full 25.6 x 25.6 km$^2$ model domain, the cloud fraction is captured much more robustly leading to a smoother daily cycle.

The two observational products provided by the LASSO library are the Total Sky Imager cloud fraction (TSI) and the low cloud fraction provided by the Active Remote Sensing of Clouds (ARSCL) value added product. These two products differ by roughly 0.2 at any moment, and when averaging over all 24 days a mean difference of roughly 0.1 remains. But the observations are clear enough to show that the MicroHH LES are substantially underestimating cloud fractions in two cases. The first is when high cloud fractions occur before noon, for example for days 20160818 and 20170627. We attribute this to the presence of non-convective clouds. Non-convective clouds should be screened out in our analysis as detailed in Section 3, so we do not expect the lack of these early clouds in the LES to affect our analysis. The second clear case of cloud underestimation in the simulations are three consecutive days in 2017 when, for reasons we do not know, the cloud fraction remains below 0.1 throughout the day (20170719, 20170720, 20170721 Figure 2).

In regards to the mean cloud fraction over all available data points after 9 am, the simulations underestimate cloud fraction by 15 % compared to the TSI cloud fraction, and by 60 % compared to the ARSCL value added product. This general underestimation is also present in the original LASSO simulations as shown by (Gustafson et al., 2020), and matches the experience of previous continental shallow cumulus LES over the ARM-SGP site (Zhang et al., 2017) and over Caubauw in the Netherlands (Schalkwijk et al., 2015). Recently Fast et al. (2019) showed that using a more realistic surface moisture distribution leads to larger and longer lived clouds, indicating that our cloud underestimation could be due to the homogeneous surface conditions of the LASSO setup. According to Gustafson et al. (2020), the simulated clouds in the LASSO simulations appear roughly 2 hours later than in the observations, which could contribute to why on average the MicroHH cloud underestimation is stronger before 3 pm (bottom left plot Figure 2). For the rest of the paper we will work with the assumption that the simulated cloud base height is accurate, and that the simulated cloud fraction is too low by an unknown amount between 10 and 60 %.

## 3 Detecting cloud chords and interpolating scenes

To avoid confusion we will first explain how we use the terms *cloud chord*, *chord base height*, and *scene* in this paper as illustrated in Figure 3. By *cloud chord* we mean a continuous 1D string of cloud. Cloud chords are also referred to as traverses (e.g. Warner, 1970b) or intercepts (e.g. Rodts et al., 2003) in airplane studies, and can theoretically fully describe the cloud field if one assumes a simplified cloud shape (Wood and Field, 2011). As we are comparing with lidar observations, our *cloud chords* are sampled from the surface, and a single cloud chord can be sampled from multiple different overlapping clouds which do not touch each other in the 3D field as hinted at by the orange chord in Figure 3. Following Lareau et al. (2018), we define the *chord base height* as the 25th percentile of the bottoms of the cloud chord grid cells. We use the term *scene* to refer to the 2D vertical slice from the surface through the cloud chord. The height of the scene is normalized from zero at the surface

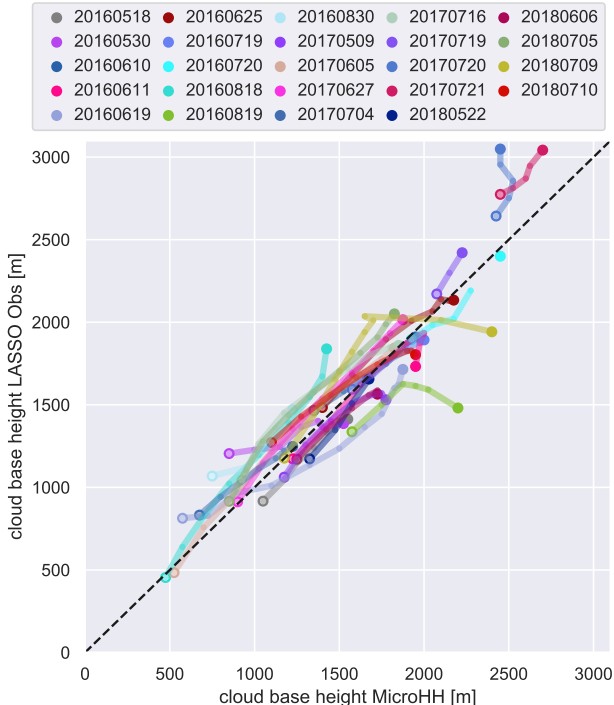

**Figure 1.** Cloud base height of the MicroHH simulations compared to the cloud base height provided by the LASSO Bundle browser, which is an ARSCL value-added data product. Points are only shown when both the model and observations have well defined cloud base at the same time, and the observations are only used when the quality flag is good. Empty circles mark the earliest time point and the large dots the latest.

to 1 at the chord base height, and extends before and behind the cloud chord as shown in the two scenes shown in Figure 3. All scenes are normalized to the same 2D grid, which allows us to merge multiple scenes together into composites. Through out all of this paper we will only sort chords according to their length, and do not sort them into forced or active (as done by Lamer and Kollias, 2015) or into updrafts and downdrafts (as done by Lareau et al., 2018).

5    In this section we detail how we define and detect cloud chords from 1D column output and 3D snapshots, followed by an analysis of how these two approaches differ. We try to be as consistent as possible with the definitions used by Lareau et al. (2018), and the exact technical implementation is described in the scripts used for this paper which are included in the supplementary material. Since the model output is not identical to the measurements used by Lareau et al. (2018) we must first define some analogue definitions. Lareau et al. (2018) detect cloudiness from a threshold back-scatter value, whereas we

10    consider a model grid box to be cloudy when the liquid water mixing ration is higher than $10^{-6}$ kg/kg. While we and Lareau et al. (2018) both only take cloudy cells that are no more than 300 m higher than the convective boundary layer (CBL) into account, we must use a different definition of convective boundary layer height. This is because the vertical velocity variance used by Lareau et al. (2018) is computed from a 2D time-height slice of vertical velocity that contains many gaps where

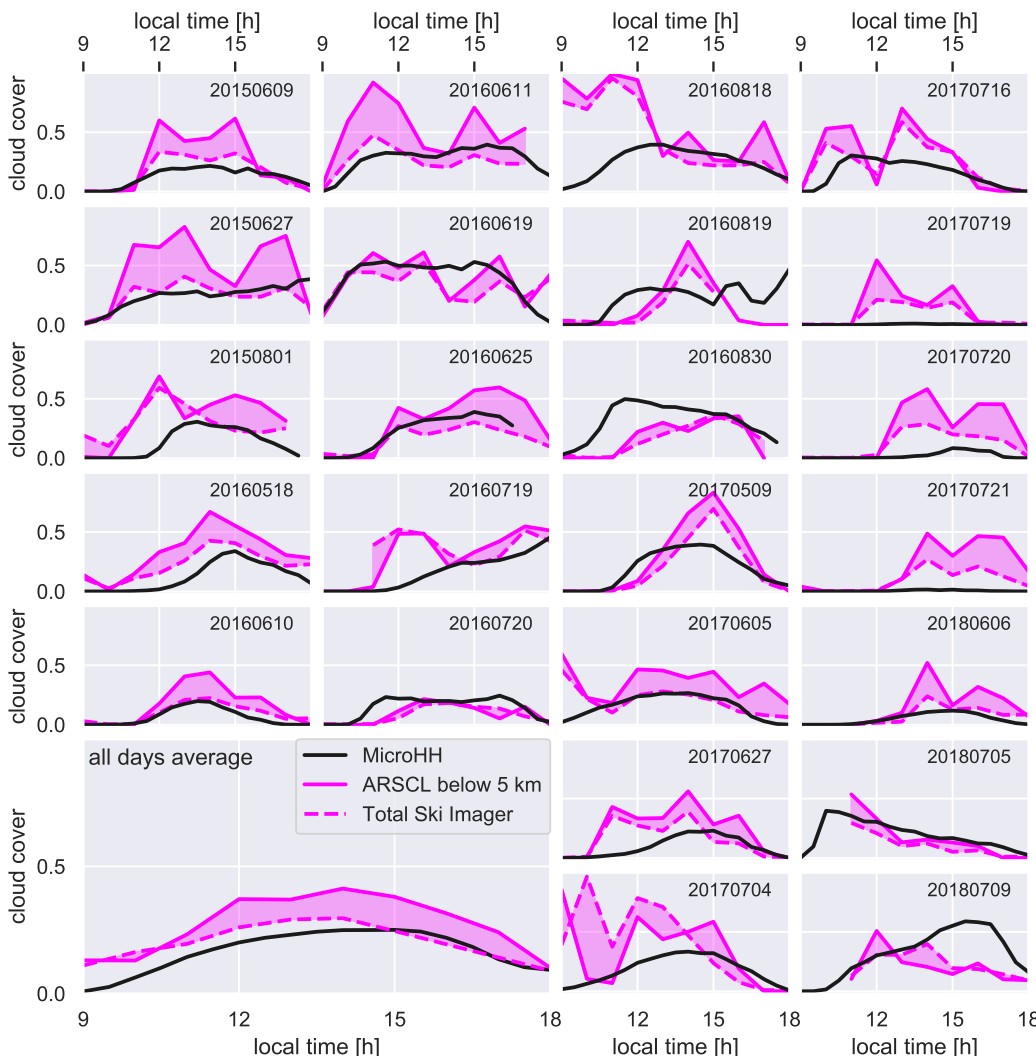

**Figure 2.** Daily evolution of two cloud fraction measurements provided hourly by the LASSO bundle browser, along with the MicroHH cloud fraction over the 25.6 x 25.6 km model domain for 24 of the 28 simulations. The remaining 4 were not included because the cloud fraction is affected by numerical artifacts at the upper model boundary. The Total Sky Imager detects cloud fraction from a fish eye image of the sky, while the ARSCL is a value added product in which cloud fraction below 5 km is determined for 15-minute windows. Observations are only used when the quality flags are ideal. The mean cloud fraction over all days is shown on the bottom left.

sufficient aerosols are unavailable to retrieve a signal or liquid water attenuates the lidar too strongly. As we can not reproduce when and where the lidar would or would not have a sufficient signal to produce a measurement, we instead use the lifting condensation level (LCL) as an additional height criteria. When the LCL is lower than the CBL height determined by the vertical velocity variance threshold, we require that the cloudy cells be no more than 300 m higher than the LCL. Given that

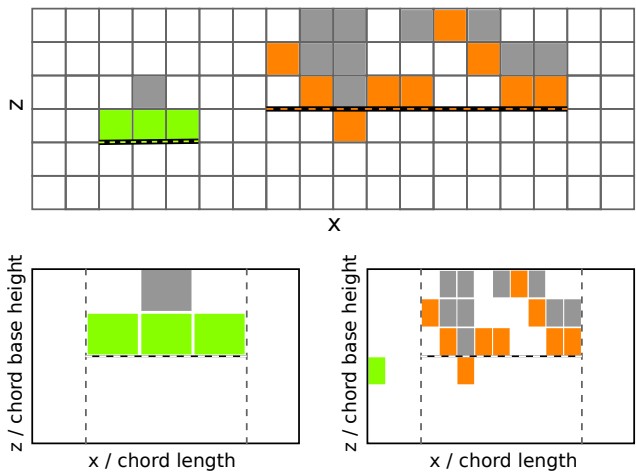

**Figure 3.** Illustration of chords, chord base height, and scenes as defined in section 3. The top half shows a sketch of a z-x slice of 3D output. The filled in grid boxes are cloudy cells. The green and orange cells show the cloudy cells which would be detected as two chords from the surface. The thick black lines mark the chord length and the chord base height of the two chords. The two lower boxes show the resulting 2D scenes after normalizing the two chords with their respective chord base height and length. For an example using Doppler lidar data see Lareau et al. (2018, Figure 1).

we do not require a high degree of precision we approximate the LCL from the mean temperature $T$ and dewpoint $T_d$ of the lowest model layer following the approximation of Espy (1836), $LCL = 125 \cdot (T - T_d)$, instead of using the exact expression (Romps, 2017).

In many ways it is simpler to detect chords from the simulation output than from the observations used by Lareau et al.
5   (2018). Firstly, we know that only convective clouds occur in the LES simulations, so none of the safeguards used by Lareau et al. (2018) to minimize the chance of accidentally sampling stratus clouds are needed. Secondly, the model output contains no gaps in data to work around. Thirdly, in contrast to the observations the horizontal and vertical wind points in the model are collocated in time and space. And finally, having the full 3D domain makes it possible to accurately split the variable into a horizontal mean and local anomaly. From the observations only the local anomaly of vertical velocity can be determined easily,
10   as we know the mean vertical velocity must be very close to zero. To calculate the observed anomalies of water vapor mixing ratio discussed in subsection 6 we use a 90 minute running mean.

### 3.1   From 1D model columns

The 1D column outputs are created by outputting the model state in specific columns of the model grid at each timestep. Each 1D column output has two dimensions, height $z$ and time $t$. Using 1D column output allows a direct comparison with the lidar
15   observations as both are stationary 1D measurements, which allows chords to be detected almost identically to the approach illustrated in Figure 1 of Lareau et al. (2018). In contrast, comparing simulations to flight observations is much more difficult because it requires imitating a planes possible flight path as discussed by Hoffmann et al. (2014).

When detecting chords from the 1D model columns we apply the same minimum chord duration of 30 seconds and maximum gap time of 20 seconds as applied to the lidar observations. But we do not apply the maximum chord duration of 20 minutes which is applied to the lidar observations. This 20 minute maximum duration is one of the safe guards implemented by Lareau et al. (2018) to reduce the chance of accidentally sampling stratus clouds, which are not present in the simulations.

It is noteworthy that as MicroHH does not have a constant time step, and accordingly the 1D column output has a constantly varying time resolution. For reference, in one simulation the mean timestep over the day is 1.2 s with a standard deviation of 0.2 s. The smallest timestep was 0.04 s, and the largest 1.6 s. On average the MicroHH timestep is very close to the 1.3 s resolution of the Doppler lidar observations.

## 3.2   From 3D model snapshots

When calculating the chords and scenes from the 1D column output and the Doppler lidar data, we use time to turn the 1D height measurements into a 2D slice with height and time as the axes. In contrast, to detect chords from the 3D output snapshots we first slice the 3D field into a 2D x-z slice and a 2D y-z slice per snapshot. From these two slices we measure the chord length by moving an imaginary Doppler lidar across the surface. The approach of cutting in x and y direction was used to detect chord length from MODIS data by Wood and Field (2011), by Endo et al. (2019) to determine $w$ across the cloud edge,

and by Sakradzija and Klingebiel (2020) to determine chord length in LES. In contrast to the chords derived from the 1D column output or the observations we are no longer measuring the duration of the chords, but the distance. We do derive a cloud duration using the mean horizontal wind speed at cloud base, but this approximation is only used as a rough comparison (shown in Figure 6).

It is noteworthy that we also replace the 20 second minimum duration threshold of Lareau et al. (2018) used for the 1D/DL chords with a 75

m threshold, which equals 3 model grid cells. Since chords shorter than 250 m have no more than 10 cells, we do not allow for gaps in the 3D cloud chords. We test in section 4.3 the impact of not allowing gaps on longer chords. When calculating composite scenes we require a minimum chord length of 10 cells (250 m) to avoid very strong resolution differences between shorter and longer scenes.

An important characteristic of our method of slicing the 3D snapshots in x and y direction is that the detected cloud chords

are not aligned with the wind direction. In contrast, the 1D and Doppler lidar chords are by nature affected by the wind direction which moves the clouds above the static sensor. Wind direction is not commonly taken into account when converting 1D measurements to cloud fractions or similar (e.g. Brooks et al., 2005; Illingworth et al., 2007), but has a noticeable influence on our spatially connected 2D scenes. If the clouds in the domain are consistently orientated in some manner to the wind direction, this could lead to differences in chord length and duration between the 3D and 1D chords. However, while we have

not systematically analyzed the orientation of the clouds in regard to wind speed, we did not notice any clear formation of cloud streets or similar in the simulations. We are confident that the various other differences in methodology between how the 1D and 3D chords are detected have a far greater effect.

While we do not take the wind direction into account when detecting chords, we try to take it into account when calculating the corresponding scenes. The method we devised to do this is by first flipping the x-z and y-z slices in accordance with the

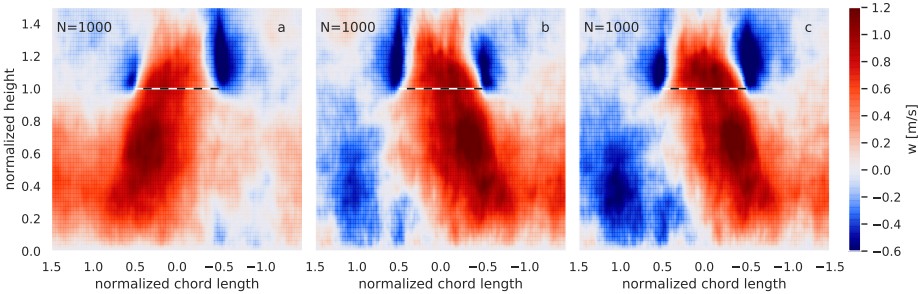

**Figure 4.** Mean vertical velocity scene of 500 chords in x-direction and 500 chords in y-direction from a 3D snapshot of the MicroHH simulation on 2016-06-11. The black dashed and white line marks the chord base height. Shown are scenes that result from not taking the wind direction into account (a), weighting the x and y scenes using the wind speed at chord base height before merging (b), and weighting the x and y scenes differently at each height using the vertical profile of horizontal wind (c). See subsection 3.2 for a detailed description of how scenes are derived from 3D snapshots as illustrated in Figure 3.

wind speed so that the imitation lidar moves along the surface in the opposite direction of the wind in x and y direction. We then calculate and store the normalized scenes from the chords in x and y direction separately for each snapshot. The last step is to weight the scenes from the x and y slices by the strength of the wind in x and y direction before merging them together to one scene. The scenes from chords with differing heights and lengths can be merged thanks to the scenes being normalized in both height and distance as shown in Figure 3.

To illustrate how the wind weighted merging works, if the wind direction were zero in y direction, then the scene resulting from merging the weighed x and y scenes would be identical to the scenes in x direction. However, if the x and y wind components were identical, the weighted and merged scene would consist to equal parts of the scenes in x and y directions. A missing but crucial detail is how we define the horizontal wind direction, for simplicity we use the mean horizontal wind direction in the 3D snapshot at each height. An example of how taking the wind direction into account affects the resulting vertical velocity scene is shown in Figure 4.

We acknowledge that there are other possible methods to detect chords, for example by moving the imitated lidar diagonally across the surface cutting through 2D clouds which would result in continuous chord lengths as done by Barron et al. (In preparation), or we could have first rotated and interpolated the 2D horizontal fields onto a 2D grid oriented to the wind direction. However, in this paper all scenes from 3D snapshots are derived using our method introduced above because it is both technically and computationally cheaper, while avoiding any artificial interpolation artefacts.

### 3.3 Differences 1D vs 3D

In comparing 1D and 3D scenes and chords it is important to recognize that the DL and 1D model data are taken from slices in time, whereas the 3D snapshots that are cut into 2D slices are a function of space. Accordingly the 1D and DL chords are sampled from clouds as they are evolving from cloud birth to death, while the 3D chords are taken from clouds frozen

**Table 2.** Number of chords diagnosed from model output, Doppler lidar (DL) and Ramen lidar (RL) observations. There are more chords then scenes in the 3D output, because we only require 3 clouds cells for a chord to be detected, but only calculate the scene of the chord if the chord contains at least 10 cells.

| 3D chords | 3D scenes | 1D column | DL | RL |
|-----------|-----------|-----------|------|-----|
| 7083946 | 4249468 | 13668 | 8132 | 778 |

in time. When the horizontal wind speeds are sufficiently low, a 1D chord could contain the complete cloud life cycle from birth to death. We can not deduce from the lidar observations alone at which stage of their life cycle the clouds are when they are sampled. Even neglecting the cloud life cycle, wind shear and rotation will stretch and deform the atmosphere as it passes over the lidar. The way we calculate scenes takes the rotation of the wind direction with height into account, but it is still a crude approximation. No post-processing method can bridge the fundamental difference between the 1D and 3D chords. Any systematic differences between the simulated 1D and 3D chords that go beyond chord definitions should be due to the difference in sampling over space or time.

## 4 Chord distributions

This section evaluates three different aspects of the cloud chords. The first evaluation looks at chord base height, mean horizontal velocity, and the time of day. The purpose of this first evaluation is to determine how similar the distribution of atmospheric conditions are of the simulated and observed chords. The second evaluation focuses on the chord length and duration. While chord base height and horizontal velocity are controlled by the prescribed LASSO forcing, chord length and duration are highly dependent on the resolved cloud geometry. And the cloud geometry in turn results form the simulated convective dynamics. The final evaluation is a sensitivity test to determine how much chord length and duration depend on the exact definition of cloudiness and if cloud gaps are permitted within a chord.

The differences between the 1D model chords and the Doppler lidar chords will reveal how close the geometry of the simulated cloud fields were to those measured at the ARM-SGP site, and the comparison of the 1D to the 3D model chords will reveal the differences between the temporal and spatial sampling discussed in section 3.3.

### 4.1 Wind, chord base height, and time of day

The histograms of horizontal wind speed and chord base height both reveal significant differences between the observed and simulated chords (Figure 5). Despite the spread being very similar the model chords are 2 m/s slower on average than the observed chords (mean 1D: 5.3 m/s, mean DL: 7.2 m/s). While there are an equal amount of Doppler lidar chords with wind speeds below 2.5 as there are above 12.5 m/s, the model chords rarely exceed 10 m/s. The chord base height distributions also

do not match well. While the observed and simulated chords both have the same maximum values at around 3000 m, there are many more model chords lower than 1500 m. On average the model chords are 230 m lower.

These differences between the observed and modelled cloud chords could be attributed to two sources. Either the simulated days are not fully representative of the atmospheric conditions of the included observations. Or the simulated days are not representative of the actual conditions at the ARM-SGP site at that time.

In regards to the cloud base, we have already shown that the simulations matched the observations at the ARM-SGP site very well (Figure 1). Despite having the correct cloud base height, the simulations could have too few clouds late in the day when clouds are higher and too many in the morning when clouds are lower. But this is not the case, because the distributions of when during the day the chords were detected show no marked shift (Subplot c, Figure 5). This absence of a shift despite the average cloud cover being lower in the simulations than the observations before noon (Figure 2), indicates that the simulated clouds are present at the right time but not as large as their observed counterparts. Note that when screening the lidar observations for chords we exclude all clouds before 10 am to avoid stratocumulus conditions.

In regards to the horizontal wind speed, while we can not easily check the horizontal wind against observations because horizontal wind is not one of the LASSO evaluation observations, we find it highly unlikely that the LASSO forcing data would contain such a strong bias. Accordingly, we believe that the discrepancies between the modelled and observed chords are not due to the simulations misrepresenting the conditions at the ARM-SGP site. From this it follows that the differences are predominantly due to the 28 days of simulations we use not being fully representative of the wind speeds and cloud heights which result from the selection method of Lareau et al. (2018) to determine which days are shallow cumulus days.

The difference in wind speed distribution could substantially effect our analysis in three ways. First, the faster a cloud is advected over lidar the less time it has to evolve during the ongoing measurement. Second, higher horizontal speeds are typically associated with higher *vertical* shear of the horizontal wind, leading to greater deformation of the scene. Last, higher wind speeds should reduce the measured cloud duration.

## 4.2 Length and duration

In this subsection we examine the differences in chord length and duration among our 3 methods of chord detection. We discuss which differences can be attributed to either the varying definitions used to determine a chord, or the model underestimation of cloud cover noted in subsection 2.1, or the difference in horizontal wind speed and height noted in the previous subsection 4.1.

We begin by comparing the duration of the Doppler lidar chords DL to the 1D output model chords (1D) (Figure 6 a). Both show a strong decrease in chord occurrence in regards to duration, with the most likely chord duration being just above the 30 seconds minimum time as applied by Lareau et al. (2018). The 1D chords can last longer than 20 minutes, given that we did not apply any maximum time measure to the model data because there was no need to filter out stratus clouds. The distribution of the 1D chords and DL chords are remarkably similar, which at first glance would indicate that the observed and simulated clouds are of the same size and shape. However, since the horizontal wind speeds are higher than in the observations (Figure 5), the chord length, which results from multiplying chord duration with the wind at cloud base, shows that longer chords are substantially more common in the observations than in the simulations (Figure 6 b). From this we can conclude

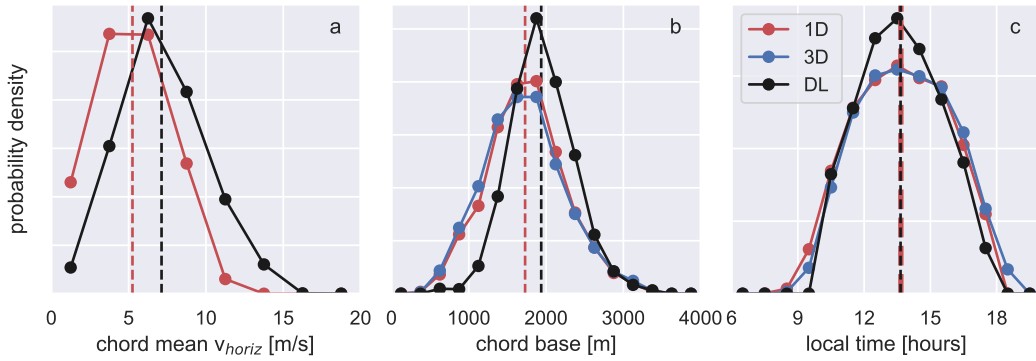

**Figure 5.** Probability density of the mean horizontal wind speed below the chord (a), chord base height (b), and time of day (c) of all chords detected from the Doppler lidar, the 1D column output, and the 3D snapshots. The dashed vertical lines mark the mean values. Since the mean time of day is almost identical (13.66 vs 13.65) the one line was plotted thicker for visibility.

that not only do the simulations have a lower cloud fraction as shown in section 2.1, the simulated clouds are also smaller than those observed. These findings fit with the results of Fast et al. (2019) that using homogeneous surface forcing in LES leads to smaller individual clouds. It is worth mentioning that the reason that the most common chord length lies at roughly 300 m is because a combination of the 30 second minimum duration and the mean horizontal wind speed being 5-7 m/s. Chords shorter than 300 m can only be measured when the horizontal wind speed is below average.

When interpreting the 3D chords one wonders if the much larger sample size leads to a higher maximum chord lengths. The maximum value is expected to increase with sample size if all samples are all drawn from the same infinite distribution. However, given that the domain is 25.6 x 25.6 km$^2$ and the simulations rarely have a cloud fraction over 0.5 (Figure 2 there is a limit on possible chord length (6 b). Note that the few very long chords in the order of 10 km are not sampled from a single 10km wide cumulus cloud, but instead are the very rare occurrences when a single chord is sampled from multiple slightly overlapping cumulus clouds.

That chord occurrence decreases strongly with increasing length and duration is hardly surprising given that it has been well established that the cumuli size density roughly follow a power law of -3 to -2 (e.g. Raga et al., 1990; Benner and Curry, 1998; Neggers et al., 2003; Zhao and Girolamo, 2007; Dawe and Austin, 2012; van Laar et al., 2019). Given that randomly cutting through a cloud with an area of $A$ creates many chords with a length smaller than $\sqrt{A}$ but only a few larger than $\sqrt{A}$ (Barron et al., In preparation), it is expected that the chord distribution is heavily dominated by small chords. Our results do not agree with the constant slope found by Wood and Field (2011) for chords up to 1000 km, as Wood and Field (2011) do not exclusively look at cumulus clouds and the satellite data they used has a maximum pixel resolution of 1 km.

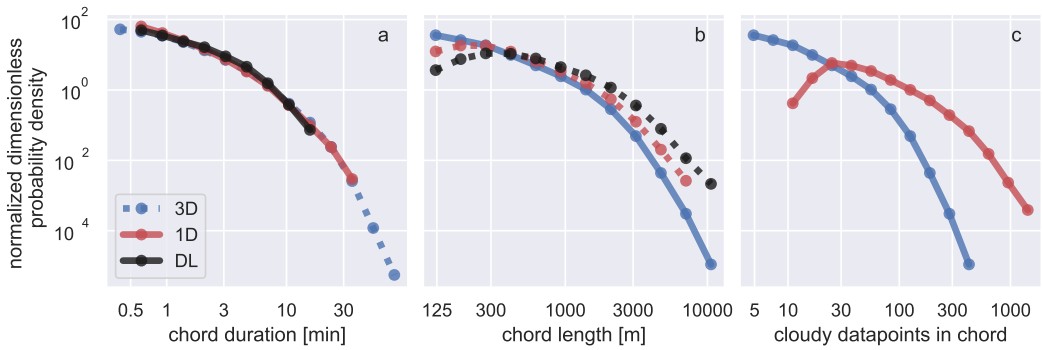

**Figure 6.** Distributions of a) cloud chord duration, b) chord length, c) number of cloudy datapoints per chord. Distributions of all chords listed in table 2 derived from the Doppler lidar (DL) observations, 3D model snapshots (3D), and 1D column output (1D) (description of chord detection in sections 3). Dashed lines indicate that the variable is derived from wind speeds.

## 4.3 Chord definition sensitivity

In this subsection we explore the sensitivity of the simulated chord distributions are to variations in the gap to variations in the gap tolerance and cloudy pixel definition used to define chords in the simulations. First, we examine the influence of allowing gaps in the 3D chords, which were not allowed in our base line results. We test this by allowing 50 m gaps (2 grid cells) and
rerunning the chord detection script for five days. Given that we are now allowing 50 m gaps we only look at chords which are at least 250 m long. While it is clear that allowing gaps will lead to more longer chords and less short chords, we can not predict how strong that impact will be. A large impact would indicate that the simulated clouds have either many small gaps in the clouds, ragged edges, or that the clouds are often only 50 m away from each other. A small impact indicates that the clouds are well isolated from each other, with few internal gaps. Our test reveals that allowing gaps has a slight but noticeable
effect on chords from 250-400 m and longer than 3000 m (Figure 7 a), indicating that the chords in between those lengths are sampled from coherent clouds.

Second, we probe the sensitivity of the chord statistics to relaxing our cloudy pixel definition by treating cells with a relative humidity of 99 or 97 % as cloudy as well. This could in theory help explain why there are more longer observed chords than detected from the 1D columns. Using a sub-saturated threshold can be justified by assuming that a volume of 25x25x25 m air
with such a high relative humidity could contain some condensed liquid, and that by requiring liquid water to be present in the model cells to count as cloudy we might be stricter than the back scatter threshold used by Lareau et al. (2018). However, by looking at the changes in the cloud duration distribution from 18 days worth of 1D column output we see that using a relative humidity has no noticeable impact on the chord duration (Figure 7). This strengthens our conclusion that more longer chords are observed because the simulated clouds are smaller. We did find that using a relative humidity slightly lowers cloud base,
but not by an amount which could impact our results.

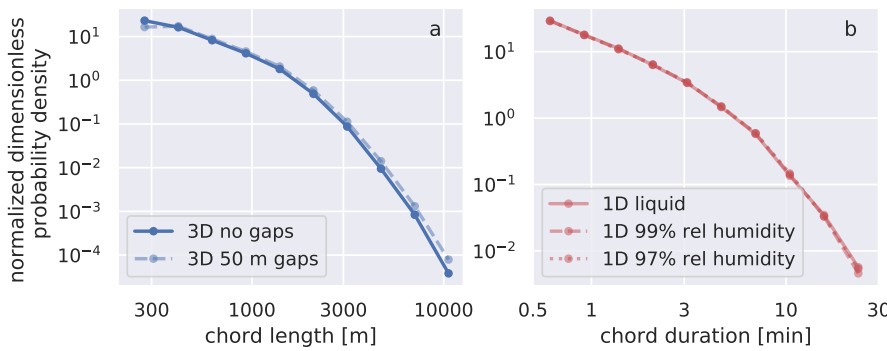

**Figure 7.** Distributions of a) chord length from 5 simulations and b) cloud chord duration from 15 simulations. The standard chord definitions introduced in section 3 were modified to test the sensitivity of the results to allowing 50 m gaps in 3D chords (a) and replacing the cloud definition with a relative humidity threshold (b).

In summary, these sensitivity tests indicate that our results are minimally sensitive to variations in the definition of cloudiness and chord continuity, indicating that the vast majority of our scenes are sampled from coherent clouds.

## 5 Chord vertical velocity

Now that we have established how the distributions of the modelled and observed chords compare, we turn to the main question of our paper. Do the vertical velocities of the modelled chords have a size dependence, and does it match that of the observations? After answering that we will explore what can be learned from investigating the axial asymmetry of the 2D scenes.

### 5.1 Size dependence

Our analysis of the size dependence of updraft strength requires some additional data stratification. First, we restrict our analysis to chords shorter than 3000 m for practical and scientific reasons. Given that the cloud base at the ARM-SGP observatory rarely surpasses 3 km (see Figure 1) it is likely that most chords longer than 3000 m are either stratus clouds that eluded the screening, or a chain of smaller cumulus clouds which by chance happen to overlap. The size filtering is useful since clouds in either of these groupings do not reflect the strength-to-size-dependence of individual cumuli.

We define the region over which to examine the mean sub-cloud vertical velocity ($w$) as a box marked in each subplot of Figure 9. The box extends vertically from 0.6 to 0.8, which we chose to capture the peak in $w$ at roughly 0.7 while staying far away enough from the cloud base to reduce the impact of the exact cloud height definition and drop contamination. The horizontal extent of the box (-0.4 to 0.4) does not quite extend all the way from chord beginning to end to reduce the possibility of subsiding shells being included. We manually defined these box dimensions, but checked that minor adjustments of height or width by $\pm\,0.05$ do not effect our qualitative results.

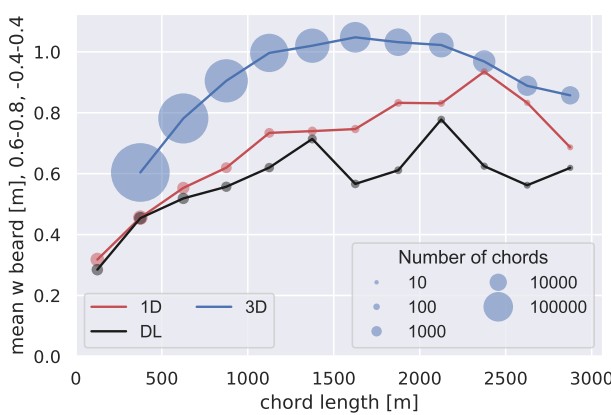

**Figure 8.** Mean normalized vertical velocity below chord base as a function of chord length. The area over which the vertical velocity is averaged is marked by the grey box in figure 9. The size of the markers represents the number of chords in that 250 m bin. Note that there is no 3D value for chords below 250 m as we only compute scenes for chords with at least 10 data points as explained in section 2.

Our representation of the subcloud velocity differs from the more commonly used $w$ at chord base, which is often considered (e.g. Endo et al., 2019) as vertical velocity at cloud base is both easily defined and an important property of mass flux parametrizations (Neggers et al., 2006; Sakradzija and Klingebiel, 2020). This differences is motivated by the sensitivity of chord base height, and thus $w$ at the chord base. For example, if the cloud base determined by lidar backscatter were on average
100 m lower than that of the cloud base detected from liquid water content in simulations, the vertical velocity at cloud base would be higher in the observations as the vertical velocity decreases with height. The second reason is due to the possibility of drops falling into the air below the cloud contaminating the retrieved signal and leading to lower $w$. We believe this effect is at least partially responsible for the very uniform $w$ decrease visible just below cloud base (see Figure 9 d and e). We do however have an analysis of $w$ at cloud base in the simulations in section 7.
We also opted to avoid using percentiles, such as the 95th percentile used by Lareau et al. (2018), despite the strongest updrafts being of critical importance to cloud formation. This is because the number of data points per chord is directly dependent on chord length for 3D chords. For example, 3D chords between 250 m and 500 m only have 10-20 data points (Figure 6 c). This increase in sampling size is also true for the 1D and DL chords to a lesser extent. Due to the large change in sampling size, calculating the mean over the 95 percentiles of chords with different lengths leads to an artificial signal.
The first result of our analysis is that the 1D and DL chords agree well for chords shorter than 1500 m (Figure 8), although a consistent bias exists within this range. The simulated chords have stronger updrafts at all lengths, with the difference between model and observation increasing to $\approx$ 0.2 m/s for chords between 1500 and 3000 m where sampling is the weakest. The bias is less than that found by Endo et al. (2019) at cloud base, which we attribute to the differences in our definition of the subcloud velocity (i.e., at cloud base vs. in the subcloud box). We also note that the $w$ PDF at cloud base studied by Endo et al. (2019)
is not sampled from cloud chords but from all cloud base grid cells, so the distinction between 1D and 3D sampling has no effect on their findings. The general shape of the 1D curve in Figure 8 is consistent with the findings of Ansmann et al. (2010)

and Lamer and Kollias (2015). A quantitative comparison is not possible since Ansmann et al. (2010) studied updrafts at a fixed height within the boundary layer, and Lamer and Kollias (2015) compare the updrafts and downdrafts of the cloud chords separately, and also normalized chord length by cloud base height and vertical velocity by the convective velocity scale ($w^*$).

While it could be argued that the 1D and DL chords have the same behaviour for chords shorter than 1500 m, the differences between the 1D and 3D chords are much more pronounced (Figure 8). The 3D chords show a stronger scaling until 1000 m, with a weak signal beyond but with a distinct small peak at 1500-1750 m.

In conclusion, 1D and DL chords are in good agreement for chords shorter than 1500 m. While a modest bias exists between 1D and DL in that range, it is significantly smaller than the increase of the mean $w$ across that range. In addition, this bias is much smaller than the difference between both 1D columns on the one hand and 3D snapshots on the other. All three chord measures show a marked increase in $w$ in the first kilometer; while the 1D model chords show a clear size dependence up to 2000 m, the 3D chords level off after 1000 m, and the signal in the observations is only beyond a doubt for the first 1000 m.

## 5.2  $w$ scene

In this subsection we examine the size and shape sensitivity of the broader subcloud vertical velocity *scene* shown in Figure 9. The analysis uses the time-height normalized composite scene for clouds in three size bins, 250-750 m (a,b,c), 750-1500 m (d,e,f), and 1500-2500 m (g,h,i).

The subplots of the left column are similar to those included in the supplementary material of Lareau et al. (2018), but contain more data and are binned differently. Note that the observed data are limited to regions where >70 % of the observations have sufficient aerosol and are cloud free, reducing the available data above 0.8 of cloud base height on the scene periphery. Since we do not know if the occurrence of sufficient aerosol at cloud base height is independent of $w$, we are not fully confident in the observed vertical velocities above 0.8 on the scene periphery.

In addition, we are wary of DL data close to cloud base (>0.9 beneath the chord) due possible interactions with cloud droplets. For example, we suspect that the reduction to 0 in vertical velocity at cloud base in Subplots a, d, and g of Figure 9 is an artifact of the cloud droplets impacting the $w$ retrieval. This is supported by the simulated scenes not having such a feature at all, and also there being no theoretical reason for updrafts to stop at cloud base.

Figure 9 indicates that the simulations and observations disagree in two aspects. Firstly, the 1D scenes have no downdrafts extending below 0.8 normalized height (Figure 9 middle column). This could indicate that the model resolution, microphysics, or 1D radiation are insufficient to fully capture the evaporative cooling at the cloud edge (Abma et al., 2013), which fits with the findings of Endo et al. (2019). Secondly, the vertical velocity is higher in the simulated chords longer than 750 m as discussed in subsection 5.1. But otherwise the 1D and DL scenes show a very similar pattern and extent. Both have stronger downdrafts in the wake of the chord, with the strongest updraft located at roughly 0.7 normalized height with a marked shift to the right, i.e. the chords have a stronger updraft earlier on. Not visible in the observations is that, as expected, the updrafts of the longer chords extend to a greater altitude.

While we can not fully isolate what causes the differences between the 1D and DL scenes, we find that the following general picture has emerged from our analysis. On average the simulated clouds are more compact and closely linked to convective

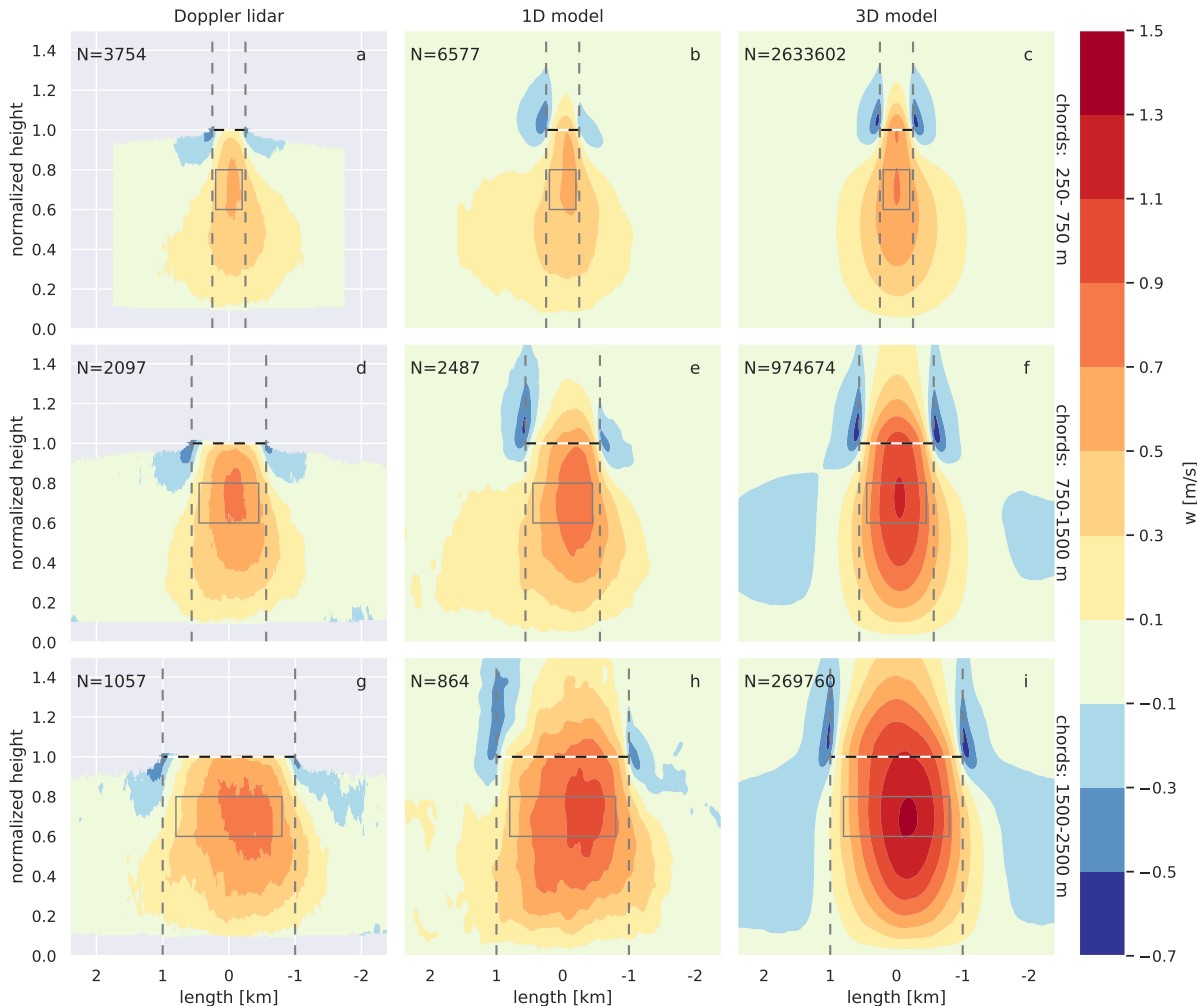

**Figure 9.** Normalized vertical velocity scenes from Doppler lidar observations (a,d,g), from 1D columns output (b,e,h), and from 3D snapshot (c,f,i). The scenes are composites of the normalized scenes binned by chord lengths of 250-750 m (a,b,c), 750-1500 m (d,e,f), and 1500-2500 m (g,h,i). After merging the scenes the normalized length is rescaled to match the mean length of the bin (e.g. 500 m for 250-750 m). The black dashed and white horizontal line marks the chord base, the dashed grey lines the chord beginning and end, and the grey box area the area averaged for figure 8. See section 2.1 for information on the Doppler data, and sections 3.2 and 3.1 for descriptions of how chords were derived from the MicroHH simulations. Doppler lidar scenes are only shown where at least 70 % of the scenes contain data.

plumes than the observed plumes. Given how small the differences are for shorter chords there are many possible explanations, such as the difference in horizontal wind shown in Figure 5, or the observations include some days with anomalously weak convection. In addition, the longer observed chords appear to have a weaker convective character (Figure 9 g), and show no clear link between chord length and updraft strength. This could indicate that some of the longest observed cloud chords are

stratus clouds that eluded our selection criteria intended to filter stratus clouds. But we can not check if this occurs at all, let alone sufficiently often to have a noticeable effect on the mean value.

In contrast to the DL and 1D scenes, the 3D chords are more symmetric with only a weak shift of the updraft core towards the start of the chord (i.e. to the right in Figure 9). The symmetry is more pronounced when looking at the two subsiding shells at the edge of cloud chords, which are equally strong and extend to the same height. There is a slight asymmetry in that the down draft region behind the chord (left in Figure 9) is a bit wider, but not to the extent Mallaun et al. (2018) detected in airplane measurements. Also, in contrast to the 1D scenes the updraft is more tightly organised below the cloud chord. This is most clearly visible when looking at how far the updrafts trail behind the chord, .i.e. to the left in Figure 9.

Returning to the differences between 1D and 3D chords introduced in subsection 3.3, the difference between 1D and 3D can be attributed to either the life cycle of the cloud evolving as it is sampled, or the upwind advection rotating the stretching the scene as it is advected over the lidar. The shear in horizontal wind speed over the boundary layer can explain why the updrafts in the 1D and DL chords are broader below the chord (Heus and Jonker, 2008). Weaker horizontal winds closer to the surface would laterally stretch the updrafts, causing the wide weak updrafts close to the surface in the 1D and DL scenes. A wind sheer above the chord base should be visible by the scene above the chord being slightly pulled toward the center of the chord. A close look at the trailing subsiding shell in the 1D chords shows that the subsiding shell bends slightly to the right over the cloud chord (Figure 9 mid column). In contrast, the subsiding shells of the 3D chords extend straight upwards. While this is only a very weak signal, it is consistent with what we would expect from a slight horizontal wind increase above the cloud base.

The asymmetry, however, can not be explained by the vertical shear of the horizontal wind. Instead, our results suggest that the asymmetry arises from the ongoing cloud life cycle. Given that the average horizontal wind speed is about 6 m/s (see Figure 5), on average we would expect the smallest clouds to age 1.5 minutes from chord beginning to end, the middle chords 3 minutes, and the longest chords 5.5 minutes (upper, middle, and lower row of Figure 9). Given that most shallow cumulus clouds live less than 20 minutes and many far shorter (Dawe and Austin, 2012; Heus and Seifert, 2013), the clouds are substantially younger at the beginning of the chords (i.e. the right in Figure 9) than on the left. We expect younger cumulus clouds to be growing with an active updraft and older clouds to be decaying and sinking. Knowing that the chords are substantially older at the end of the chord, this ageing provides a consistent explanation why updrafts both below and above the chord base are stronger at the beginning of the cloud chords.

Having considered possible explanations for the differences in width and asymmetry between the 1D and DL chords, we have so far not explained why the 3D chords have stronger updrafts. The difference in updraft strength is very likely partially due to the 3D chords not allowing any gaps in the chords. While we have shown that allowing 50 m gaps does not effect our results (subsection 4.3), we know that allowing longer gaps over hundreds of meters does reduce the updraft velocity. Interestingly, not only are the 3D updrafts stronger, the downdrafts behind and before the chords are also stronger for the biggest chords (Figure 9 i). Given that the long chords have a strong updraft, due to mass conservation in the LES domain there must be downdrafts in between the chords. That these downdrafts are not visible in the 1D chords indicates that while the 3D

chords taken from the frozen snapshots have a clear separation of updrafts beneath the clouds and subsidence in between, this clear separation is blurred by the ongoing time evolution in the 1D chords.

## 6 Moisture anomalies

After comparing the simulated vertical velocities against the Doppler lidar measurements, we now move on to the moisture anomalies computed from the Raman lidar. Our core question remains to what degree model and observations agree, and to see if the anomalies show a clear size dependence.

For interpreting the Raman lidar results it is important to note that the sample size and frequency of the moisture anomalies is substantially lower than that of the vertical velocity. We have roughly a factor of 10 less observed chords (Table 2) and the sample frequency is also lower by a factor of 10 (10 s vs 1.3 s). To make use of all the data we have available for the largest chords we expanded the binning of the scenes shown in Figure 10 compared to Figure 9 (1500-3000 m vs 1500-2500 m). While the Raman lidar observations tend to extend higher than the CBL, they can not penetrate into the clouds (Lareau, 2020). Since we calculate the anomalies from a 90 minute running mean we do not plot the anomalies higher than 0.95 of the chord base because the missing observations in the cloud bias the resulting anomalies.

As expected, due to the sharp decrease in water vapor in the atmosphere above the boundary layer both observed and simulated moisture anomalies have a very strong maximum at cloud chord base and, for the simulations, in the cloud cores (Figure 10). In both the observations and simulations the positive moisture anomaly reaches all the way to surface, with a smooth and monotonic increase from the surface to cloud base. In our simulations the water vapor values in the lower mixed layer lie roughly between 10 and 15 g/kg, so the mean anomalies at 0.5 chord base height are on the order of 1-2 %. To avoid the signal being dominated by possible small shifts in cloud height, we use a larger and lower averaging box (marked in grey in Figure 10) to calculate the moisture anomalies shown in Figure 11 than we did for the vertical velocities (marked in grey in Figure 9).

In contrast to the vertical velocity which turns negative within the cloud edge (Figure 9), the water vapor anomalies remain slightly positive beyond the cloud at the cloud base height. This is consistent with our general understanding of subsiding shells which, while drier than the cloud cores, are still moister than the surrounding air (Heus and Jonker, 2008; Wang and Geerts, 2010; Katzwinkel et al., 2014; Lareau, 2020).

Comparing the RL and 1D moisture anomaly scenes to the 3D scenes results in the same conclusions we deduced from the vertical velocity scenes. The RL and 1D chords have a clear asymmetry with no clear negative anomalies before and behind the chords, while the 3D chords are almost symmetric with clear dry areas before and behind the chords (Figure 10).

The size dependence of the moisture anomalies below the chord base of the 1D and 3d scenes is also quite similar to that of the vertical velocity (Figure 11). Again the 3D scenes have stronger anomalies than the 1D scenes, with the highest anomalies corresponding to chords which roughly have the length of the mean boundary layer height (1.5km), while the 1D scene anomalies slowly increase with chord length till about 2.5 km. In contrast to the simulated scenes the observed moisture anomalies decrease with length, although large fluctuations between bins exists due to the small sample sizes. It should be

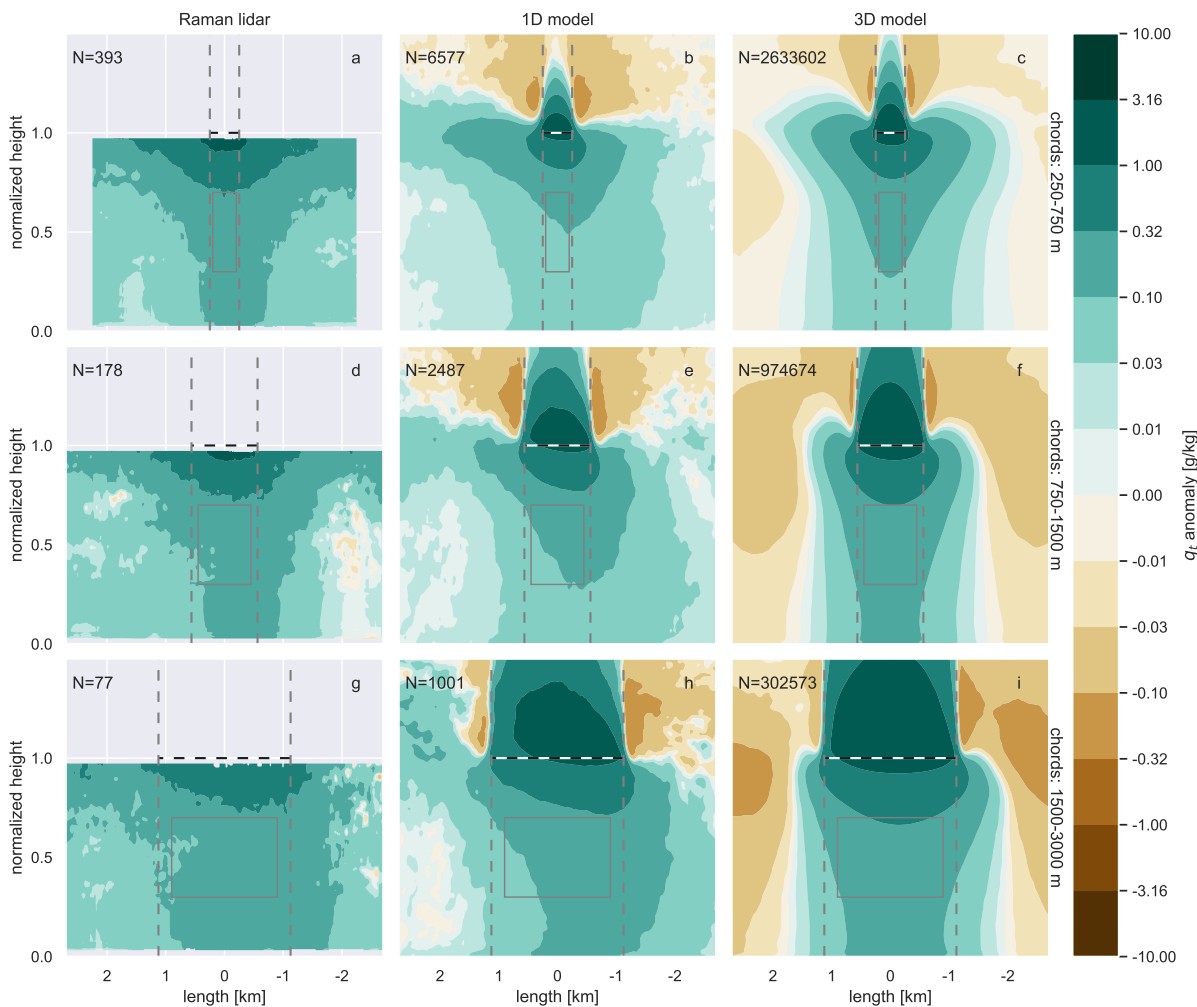

**Figure 10.** Scenes of $q_t$ anomalies in g/kg from Raman lidar observations, 1D column output, and 3d column output. The scenes are composites of the normalized scenes binned by chord lengths of 250-750 m (a,b,c), 750-1500 m (d,e,f), and 1500-2500 m (g,h,i). After merging the scenes the normalized length is rescaled to match the mean length of the bin (e.g. 500 m for 250-750 m). The black dashed and white horizontal line marks the chord base, the dashed grey lines the chord beginning and end. See section 2 for information how the scenes are computed. Note the logarithmic color bar. For reference, the water vapor values in the lower mixed layer range from 10 to 15 g/kg.

noted that the relative anomalies are only in the range of 1-2 %, and that any bias or error in the anomaly calculation could substantially effect the results. To more completely understand the size dependence in the observations would likely require substantially more data, preferably with a higher measurement frequency, and a more careful separation of anomalies from the mean than what we use. However, for now we are confident that the decreasing moisture anomalies with length are a direct result of using a running mean to calculate the anomalies.

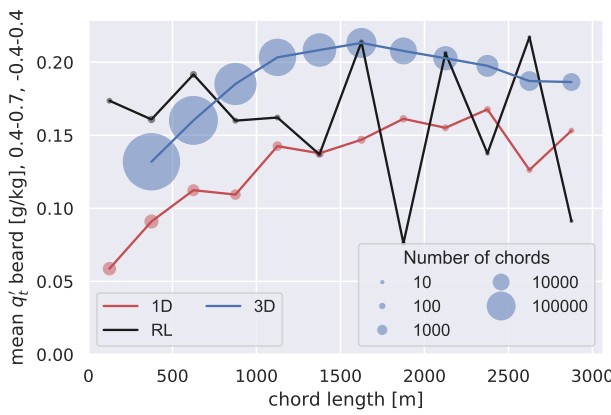

**Figure 11.** Mean water vapor anomalies below chord base as a function of chord length. The area over which the anomaly is averaged is marked by the grey box in figure 10. The size of the markers represents the number of chords in that 250 m bin. Note that there is no 3D value for chords below 250 m as we only compute scenes for chords with at least 10 data points as explained in section 2.

In summary, the moisture anomaly analysis shows that the simulations match our theoretical expectations while correctly reproducing the observations. The differences between the 1D and 3D scenes are consistent with our findings from the vertical velocity, in that the 1D chords have a weaker signal with asymmetric scenes. Both 1D and 3D scenes show a clear size dependence, while the observed chords have a weak inverse size dependence due to how the anomalies were calculated. In relative terms the anomalies below halfway below the chords are in the range of 1-2 %.

## 7 Sampling uncertainty

In this section we use simulated data to address the question how the uncertainty in cumulus properties depends on how many days of shallow cumulus lidar observations are available. Assuming a single 1D column output of a shallow cumulus simulation is equivalent to having one lidar measuring over one day, this analysis will determine how many days of shallow cumulus lidar observations would be needed at the ARM-SGP site to derive the various chord properties to within a threshold uncertainty. This analysis is based on the premise that each individual column output is both independent and equivalent, i.e. we make no distinction between two column outputs from the same day or different days. On average 31 chords are detected from each column, with a standard deviation of 17 and a maximum of 68. For this analysis we exclude all columns with less than 10 chords, leaving us with 386 columns of our total 448. The three properties that will be investigated are the chord length distribution, the mean vertical velocity below the chords as a function of chord length, and the mean $w$ scene.

The sampling uncertainty of the chord length distribution for $n$ columns is determined by repeatedly sampling $n$ columns at random and calculating the chord distribution of all the chords contained in those $n$ columns. As expected, uncertainty in the cord length distribution is directly linked to chord occurrence, with the smallest uncertainty being at about 300 m which is the most common chord length (Figure 12 a). At this point it is informative to compare these results with the DL observations. The

line presenting the DL chords roughly lies within the $n = 64$ shading, which implies that with less than 64 columns we can not rule out that the differences between the observed and simulated chord length distributions are due to sampling.

The mean $w$ at cloud base is displayed in the same manner as the chord length distribution (Figure 12 b). In contrast to chord length, the uncertainty regarding the dependence of mean $w$ at chord base on chord length is relatively constant for chords from 100 to 3000 m. We speculate that two compensating effects are behind this behavior; while longer chords are rarer than shorter chords, their mean $w$ also has a much smaller spread. Accordingly, fewer long chords are needed to reach a representative mean value. An interesting result not directly related to sample sizes is that the shortest chords below 200 m in length actually have a negative vertical velocity on average. This negative velocity for the smallest chords was also found by Rodts et al. (2003) in airplane measurements, and indicates the smallest chords are dominated by chords sampled from the subsiding shells of larger clouds or from dying clouds at the end of their life cycle.

The final variable we examine is the 2D $w$ scene. To determine the uncertainty in the $w$ scene we calculate the root mean square error (RMSE) between two randomly sampled scene composites through a bootstrapping approach. The randomly sampled scenes are generated by drawing $2 \cdot n$ of columns into two separate groups with $n$ columns each from our total pool of 386 1D columns (again we only include columns which contain at least 10 chords). Once a column is drawn it is removed from the pool so that no column is compared against itself. We then form a composite of all $w$ scenes present in the $n$ columns in each group. And finally we calculate the RMSE between the two $w$ scenes that were calculated from the two pots. This random drawing is repeated 100 times. Note that the selection of 100 permutations is arbitrary, but results remain unchanged for higher numbers.

As each column contains a varying amount of scenes with different lengths and from varying times of day, there is no easy way to predict how the RMSE will behave. Our analysis shows that on average the RMSE behaves as if drawing from normally distributed errors with a standard deviation of 0.3 m/s (Figure 13). Accordingly, quadrupling the columns included in the analysis halves the sampling uncertainty. For example, to reduce the expected sampling uncertainty from 0.05 m/s to 0.025 m/s requires measuring 128 days instead of 32.

# 8 Conclusions and discussion

The two main goals of our study were to evaluate the LES concerning (thermo)dynamic perturbations of continental transient shallow cumuli, and to establish if size dependence exists in these features. These goals were achieved by comparing cloud chords observed by Doppler and Raman lidars against chords derived from 1D and 3D model output. From our results we established the following:

## 8.1 Conclusions

- We are the first to show that the LES do reproduce the magnitude and shape of vertical velocities and moisture anomalies observed bellow shallow cumulus clouds.

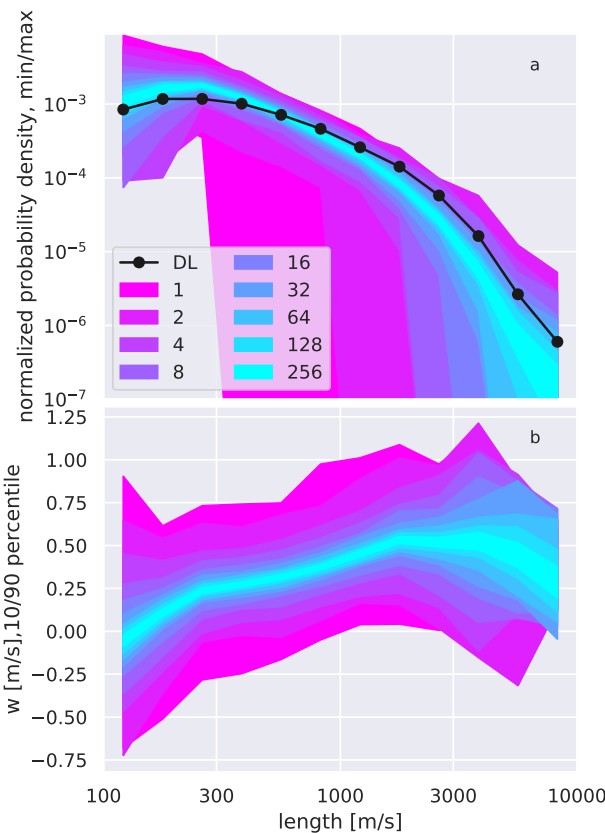

**Figure 12.** Chord length distribution (a) and mean vertical velocity below the chord binned by chord length (b) as a function of chord length, based on simulation data. The shading in the top subplot shows the minimum or maximum value that result from randomly sampling and combining $n$ columns 386 times (details in section 7). The shading and legend marks how many $n$ 1D column outputs were combined together in each random sample. The shading in the bottom figure fills the 10th till 90th percentile of all the samples. The black line is the Doppler lidar data, which has roughly the same amount of chords as we detect from 250 column outputs.

- – The size dependence of vertical velocity with chord length is clear and robust. Our work puts the findings of Rodts et al. (2003); Lamer and Kollias (2015); Neggers (2015); Lareau et al. (2018) on a more robust statistical foundation.

- – Compared to observations, MicroHH in combination with the LASSO forcing leads to lower cloud fraction (similar to Schalkwijk et al., 2015; Zhang et al., 2017; Gustafson et al., 2020) and slightly shorter cloud chords.

- – We see a positive bias in modelled vertical velocity at roughly 0.7 chord height that is smaller than 0.1 m/s for chords shorter than 1500 m. Our bias is an order of magnitude smaller than the bias noted by Endo et al. (2019) at cloud base.

- – Differences between the observations and simulations are smaller than the differences between the 1D and 3D sampling approaches applied to the simulations.

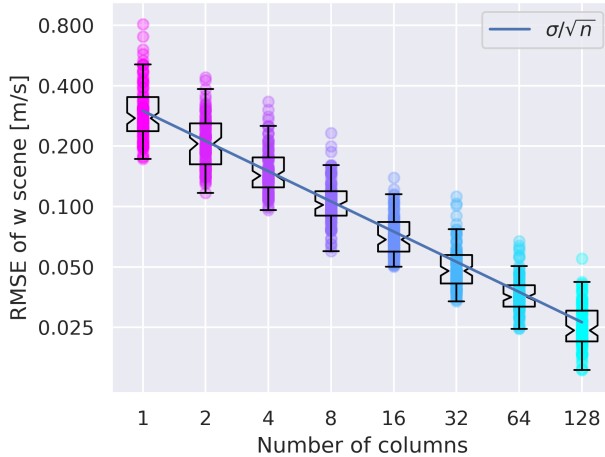

**Figure 13.** Root mean square error (RMSE) between two randomly sampled $w$ scene composites. The exact process by which the sampling occurs is explained in subsection 7, and is applied 100 times for each color. The x-axis denotes the number of 1D column outputs included in the randomly sampled scenes. Note the logarithmic y axis. The added line marks the sampling error associated with drawing $n$ samples from a normal distribution with a standard deviation of 0.3 m/s.

- – The quantitative analysis of the observed moisture anomalies dependence on chord size is complicated by the relatively weak anomalies, the need to determine the mean moisture, and the smaller data amounts available from the Raman lidar. Solving this problem is for now considered a future research effort.

- – Our results suggest that the asymmetry in the observed Doppler lidar scenes mainly originates from the cloud evolving while it is being sampled.

- – Roughly 60 days of Doppler lidar observations from the ARM-SGP site are needed to detect the differences between observed and simulated chord-length distribution with certainty.

## 8.2  Discussion

Our success at using measurements to evaluate LES in combination with using LES to better understand the measurements at the ARM-SGP site underlines the usefulness of initiatives to run continuous LES at super-sites such as LASSO (Gustafson et al., 2020) and the JOYCE testbed (Neggers et al., 2012; van Laar et al., 2019) to drive scientific progress. Our methodology also highlights again just how crucial it is to imitate the observations as closely as possible when evaluating LES against observations, and illustrates one of the many meaningful ways LES can be analyzed beyond looking at profiles of fluxes and mean variables. The underestimation of cloud fraction and cloud size in the simulations match the recent findings of Fast et al. (2019) that using a horizontally homogeneous surface moisture reduces cloud size and lifetime.

Our composites of millions of scenes taken from the 3D snapshots indicate that on average the shape of a standard symmetric and cylindrical convective plume as originally formulated by Simpson and Wiggert (1969) is a reasonable approximation,

despite individual clouds being asymmetric (Zhao and Austin, 2005) or composed of a collection of thermal bubbles (French et al., 1999; Yano, 2014). The clear scaling in vertical velocity below the clouds supports the use of multiple plumes with different sizes as originally proposed by Arakawa and Schubert (1974). Also in regards to parameterized plumes, that the moisture anomalies below the clouds are relatively small (1-2 %) indicates that initializing multiple plumes at the surface using

the background moisture is justifiable (Park, 2014; Neggers, 2015; Hagos et al., 2018).

While we expect our results from the ARM-SGP site apply qualitatively to continental shallow cumulus elsewhere, the applicability to cumulus convection in different climate regimes (e.g. marine trade-wind, mixed-phase high-latitude) and in different modes (precipitating, deep) still needs to be demonstrated. Now that we have shown that the LES capture the observed vertical velocity and moisture structure of shallow cumulus chords at the ARM-SGP site, we can confidently use the LES to

study features linked to shallow cumulus dynamics, such as convective plumes, subsiding shells, and cold pools.

*Code and data availability.*  The simulation and lidar data shown in the figures are freely available at 10.5281/zenodo.3731944, which also contains the python files to plot the data and post-process the simulations. The simulations were generated with version 1.9.1 of MicroHH https://github.com/microhh/microhh2/releases/tag/1.9.1. The data used to force MicroHH and to evaluate the simulated cloud fraction and base are available through the LASSO bundle browser https://adc.arm.gov/lassobrowser. The lidar data which was processed is freely avail-

able through the ARM data archive https://adc.arm.gov/ (see subsection 2.1).

*Author contributions.*  All authors contributed substantially to the research direction and scope, paper structure, and conclusions over a 2 year time frame. Philipp Griewank wrote the vast majority of the text, post-processed the simulations, and plotted all the figures. Thijs Heus set up and ran the simulations. Neil Lareau gathered and post-processed the observations, and helped proofread. Roel Neggers provided experience and context to interpret the results in regards to cumulus parametrizations and LES at meteorological super-sites.

*Competing interests.*  No competing interests are present.

*Acknowledgements.*  This research was supported primarily by the U.S. Department of Energy's Atmospheric System Research, an Office of Science Biological and Environmental Research program, under grant DE-SC0017999 awarded to Thijs Heus and grant DE-SC0019124 awarded to Neil Lareau. Our research would not have been possible without the ground work laid by the LASSO project (LES ARM Symbiotic Simulation and Observation) and the Southern Great Plains (ARM-SGP) atmospheric observatory established by the Atmospheric

Radiation Measurement (ARM) user facility. We would like to thank the University of Cologne for funding a mobility grant used by Philipp Griewank to visit Thijs Heus and Neil Lareau. The computing resources to run the simulations were provided by a Cleveland State University Faculty Research and Development award.

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
