# Peer review of "Size-dependence in chord characteristics from simulated and observed continental shallow cumulus"

_Atmospheric Chemistry and Physics, 2020_

## Referee Comment (RC1) · Anonymous Referee #1 · 25 May 2020

This study focuses on the properties of vertical velocity and moisture anomaly fields around shallow cumulus clouds based on large-eddy simulations and long-term lidar observations. Doppler and Raman lidars from 28 LASSO days are used to obtain the chord properties (including duration, length, and height) as well as vertical velocity and water vapor mixing ratio below the chord. The observed statistical properties of cloud chords are compared with modeling results of 1D column output and 3D snapshot based on MicroHH, in which the large-eddy simulations are applied to run all 28 cumulus days using LASSO forcing data. The motivation of this study is to determine how the amplitude and shape of the vertical velocity and moisture anomaly fields of cumulus clouds change with size and to determine if the LES simulations provide a reliable

approximation of shallow cumulus cloud statistics. Differences and similarities among lidar observations, cloud chords from 1D column and 3D snapshot are discussed thoroughly in the manuscript. A scaling of vertical velocity and moisture anomalies below the cloud chords with chord size has been found. In summary, the merit of this study is the analysis of cloud chord properties based on a large amount of observational and modeling results. However, some conclusions should be clarified/justified. Besides, grammar and spelling should be improved. My major and technical comments are listed below and should be addressed properly before the manuscript is suitable for publication in ACP.

1. Page 5, Lines 18-20 and Figure 2: Explain the reason why there are lots of cases with the cloud fraction of 0 from MicroHH. The cloud fraction from MicroHH is the mean cloud fraction from the whole field or from 15-minute windows from one point or multiple points? It is not clear to me how to get the conclusions that "with no clear temporal evolution visible" and "This strong scattering is not evident in the model simulations". Where is the "temporal information" in Figure 2? All data are scattered in my point of view. It is hard to tell whether "this strong scattering is not evident in the model simulations".

2. Page 6, Line 9 and Figure 5c: "this is at least partially due to the model spin up causing the simulated clouds to appear roughly 2 hours later than in the observations." If this is true, how to explain the results in Figure 5c in which the distributions of when during the day the chords were detected show no marked shift.

3. Page 8, Line 15: "we use the LCL". Use the LCL for what? To choose cloudy cells that are no more than 300 m higher than LCL? Please also describe how to calculate LCL in this study.

4. section 3.1: At the beginning of this section "The 1D column outputs is created by outputting the model state in specific columns of the model grid at each timestep." It reads like this subsection only focuses on modeling criteria. However, the second

paragraph focuses on the observational criteria. "we use a mix of the vertical velocity variance and the lifting condensation level" is the criteria for cloud chord detected from observation? Consider modifying the title and content in section 3.1 to make sure they are consistent and easy to read.

5. Page 10, Line 16 and Figures 9&10: "merging them together". All cloud chords are binned in three categories in Figures 9 and 10: 250-750 m, 750-1500 m, 1500-2500/3000 m. The x coordinate is the length of the chord. I understand the "merging" in y coordinate is by normalization, however, It is not clear to me how to merge them together in x coordinate if they have different chord lengths in each category.

6. Page 16, Line 6 and Figures 9&10: Clearly describe the method to choose the box in text, especially the four boundaries.

Technical comments:

1. Page 1, Line 14: "aspects" -> "aspect"

2. Page 1, Line 21: not a complete sentence.

3. Page 2, Line 2: "then" -> "than"

4. Page 3, Line 8: "provide" -> "provides"

5. Page 4, Line 27: "and are run using..." check grammar

6. Page 5, Line 1: "25.6 km2" -> "25.6 x 25.6 km$^2$"

7. Page 5, Line 2: "15 0m" -> "150 m"

8. Page 5, Line 20: "km" -> "km$^2$"

9. Page 6, Line 11: This paragraph has only one sentence.

10. Page 8, Line 8: "10-6" -> "10$^{-6}$"

11. Page 8, Line 12: "available" -> "unavailable"?

12. Page 9, Line 7: "is" -> "are"

13. Page 10, Line 10: "are confident that the various other differences in methodology between the how 1D and 3D chords are detected have a far greater effect." Check grammar.

14. Page 14, Line 14: "the the" -> "the"

15. Page 16, Line 11: "For example if the cloud base determined by the lidar were 100 m too low the vertical velocity determined would be higher." This sentence is not clear to me.

16. Page 19, Line 5: "have have" -> "have"

17. Page 19, Line 14: "at the how far. . ." -> "at how far"

18. Page 20, Line 13: "chords chords" -> "cloud chords"

––––––––––––––––––––––––––––––

---

## Referee Comment (RC2) · Anonymous Referee #2 · 2 Jun 2020

This paper compares the cloud size (chord length), under-cloud vertical velocity and moisture anomalies observed and simulated by large-eddy simulation models at ARM SGP and analyzes the cloud size dependence of vertical velocity and moisture anomalies. I think the methodology is sound and the presentation is mostly clear. I suggest minor revision. Here are my comments,

1. Please make the English more readable in terms of both grammar and style.

2. I don't quite understand Section 7 and the last bullet point in the conclusion (Sect. 8.2). How is the "30 days" conclusion reached? Also, "The line presenting the DL chords roughly lies within the n = 64 shading, at least 64 columns are needed for ruling

out sampling being the cause for the difference between observations and simulations."
Why?

3. Did Endo et al. (2019) use a methodology more similar to the 1D or 3D method in
this paper? Do you see a change in the PDF shape from 1D to 3D method? Since
Endo et al. (2019) is so recent and relevant, I suggest more discussions comparing
your results to theirs.
* * *

---

## Author Comment (AC1) · 1 Jul 2020

Everything is contained in the attached pdf.

Please also note the supplement to this comment:
https://www.atmos-chem-phys-discuss.net/acp-2020-338/acp-2020-338-AC1-supplement.pdf
* * *

---

## Author Response (AR1)

**Revisions of Paper acp-2020-338, Size-dependence in chord characteristics from simulated and observed continental shallow cumulus.**

Griewank          Heus          Lareau          Neggers

July 1, 2020

We would like to thank both reviewers for taking the time to review our paper, and are encouraged by their assessment that the paper is suited for publication in ACP after some revisions. This document contains a response to all of the points raised by the reviewers with the resulting changes to the manuscript listed. The revised manuscript contains many more language changes than mentioned here, as both reviewers noticed that the grammar and spelling could be improved. In addition we added a few more references we were made aware of. All changes to the manuscript are listed at the end of the document after respoding to the reviewer comments.

**Comments by Reviewer #1**

> Page 5, Lines 18-20 and Figure 2: Explain the reason why there are lots of cases with the cloud fraction of 0 from MicroHH. The cloud fraction from MicroHH is the mean cloud fraction from the whole field or from 15-minute windows from one point or multiple points? It is not clear to me how to get the conclusions that "with no clear temporal evolution visible" and \This strong scattering is not evident in the model simulations". Where is the "temporal information" in Figure 2? All data are scattered in my point of view. It is hard to tell whether "this strong scattering is not evident in the model simulations".

In response to this comment we decided to completely revise and expand the plot comparing the simulated and observed cloud fractions, and also slightly expanded the discussion in the paper.

Old plot:

[Figure]

New plot:

[Figure]

In contrast to cloud base height, the modelled and observed cloud fraction align less well (Figure 2). The observed hourly cloud fraction has a higher temporal variability, with measurement-to-measurement changes of up to 0.5 (e.g. 20170716 Figure 2). Such strong shifts likely represent the sampling bias in a spatially and temporally heterogeneous cumulus topped boundary layer (Rossow, 1989). Since the cloud fraction from the MicroHH simulations is calculated from 3D snapshots of the full 25.6 x 25.6 km$^2$ model domain, the cloud fraction is captured much more robustly leading to a smoother daily cycle.

The two observational products provided by the LASSO library are the Total Sky Imager cloud fraction (TSI) and the low cloud fraction provided by the Active Remote Sensing of Clouds (ARSCL) value added product. These two products differ by roughly 0.2 at any moment, and when averaging over all 24 days a mean difference of roughly 0.1 remains. But the observations are clear enough to show that the MicroHH LES are substantially underestimating cloud fractions in two cases. The first is when high cloud fractions occur before noon, for example for days 20160818 and 20170627. We attribute this

*to the presence of non-convective clouds. Non-convective clouds should be screened out in our analysis as detailed in Section 3, so we do nor expect the lack of these early clouds in the LES to affect our analysis. The second clear case of cloud underestimation in the simulations are three consecutive days in 2017 when, for reasons we do not know, the cloud fraction remains below 0.1 throughout the day (20170719, 20170720, 20170721 Figure 2).*

> Page 6, Line 9 and Figure 5c:  "this is at least partially due to the model
> spin up causing the simulated clouds to appear roughly 2 hours later than in
> the observations.  "If this is true, how to explain the results in Figure 5c
> in which the distributions of when during the day the chords were detected
> show no marked shift.

This is an important question which we neither addressed nor mentioned in the original manuscript. We now address it in the revised discussion of the cloud fraction comparison:

*The first is when high cloud fractions occur before noon, for example for days 20160818 and 20170627. We attribute this to the presence of non-convective clouds. Non-convective clouds should be screened out in our analysis as detailed in Section 3, so we do nor expect the lack of these early clouds in the LES to affect our analysis.*

And we revisit this point later in the manuscript when daily distribution of the chords is discussed:

*This absence of a shift despite the average cloud cover being lower in the simulations than the observations before noon (Figure 2), indicates that the simulated clouds are present at the right time but not as large as their observed counterparts. Note that when screening the lidar observations for chords we exclude all clouds before 10 am to avoid stratocumulus conditions.*

> Page 8, Line 15:  "we use the LCL".  Use the LCL for what?  To choose cloudy
> cells that are no more than 300 m higher than LCL? Please also describe how to
> calculate LCL in this study.

We have clarified how we use the LCL and how we calculate it.

*As we can not reproduce when and where the lidar would or would not have a sufficient signal to produce a measurement, we instead use the lifting condensation level (LCL) as an additional height criteria. When the LCL is lower than the CBL height determined by the vertical velocity variance threshold, we require that the cloudy cells be no more than 300 m higher than the LCL. Given that we do not require a high degree of precision we approximate the LCL from the mean temperature $T$ and dewpoint $T_d$ of the lowest model layer using Espy's equation, $LCL = 125 \cdot (T - T_d)$, instead of using the exact expression (Romps, 2017).*

> section 3.1:  At the beginning of this section "The 1D column outputs is
> created by outputting the model state in specific columns of the model grid
> at each time step." It reads like this subsection only focuses on modeling
> criteria.  However, the second paragraph focuses on the observational
> criteria.  "we use a mix of the vertical velocity variance and the lifting
> condensation level" is the criteria for cloud chord detected from observation?
> Consider modifying the title and content in section 3.1 to make sure they are
> consistent and easy to read.

We have changed the subsection name to "From 1D model columns" to stress that we are addressing how to derive chords from the model data, and have moved the details of how we use the LCL to the previous subsection to make the manuscript less confusing.

> Page 10, Line 16 and Figures 9 & 10:  "merging them together".  All cloud
> chords are binned in three categories in Figures 9 and 10:  250-750 m,
> 750-1500 m, 1500-2500/3000 m.  The x coordinate is the length of the chord.
> I understand the "merging" in y coordinate is by normalization, however, It
> is not clear to me how to merge them together in x coordinate if they have
> different chord lengths in each category

The scenes of differing chords can be merged because all scenes are normalized in both length (x coordinate) and height, which we now emphasize by adding the following line to what use to be Page 10, Line 16:

*The scenes from chords with differing heights and lengths can be merged thanks to the scenes being normalized in both height and distance as shown in Figure 3.*

The lengths shown in Figures 9 & 10 are achieved by first merging the normalized scenes, and then rescaling the normalized length to match the mean chord length used to bin the scenes. We added the following description to the captions of Figures 9 and 10:

*The scenes are composites of the normalized scenes binned by chord lengths of 250-750 m (a,b,c), 750-1500 m (d,e,f), and 1500-2500 m (g,h,i). After merging the scenes the normalized length is rescaled to match the mean length of the bin (e.g. 500 m for 250-750 m).*

> Page 16, Line 6 and Figures 9 & 10:  Clearly describe the method to choose the
> box in text, especially the four boundaries.

We have included a description of why we chose the used boundaries. Both for the $w$ scenes:

*We define the region over which to examine the mean sub-cloud vertical velocity (w) as a box marked in each subplot of Figure 9. The box extends vertically from 0.6 to 0.8, which we chose to capture the peak in w at roughly 0.7 while staying far away enough from the cloud base to reduce the impact of the exact cloud height definition and drop contamination. The horizontal extent of the box (-0.4 to 0.4) does not quite extend all the way from chord beginning to end to reduce the possibility of subsiding shells being included. We manually defined these box dimensions, but checked that minor adjustments of height or width by ± 0.05 do not effect our qualitative results.*

And for the $q_t$ scenes:

*To avoid the signal being dominated by possible small shifts in cloud height, we use a larger and lower averaging box (marked in grey in Figure 10) to calculate the moisture anomalies shown in Figure 11 than we did for the vertical velocities (marked in grey in Figure 9).*

> Technical comments

Thank you for these helpful comments, all were addressed.

**Comments by Reviewer #2**

> Please make the English more readable in terms of both grammar and style.

We have reworked the text substantially. The changes are marked below after responding to the reviewer comments.

> don't quite understand Section 7 and the last bullet point in the conclusion
> (Sect.8.2).  How is the "30 days" conclusion reached?  Also, "The line
> presenting the DL chords roughly lies within the n = 64 shading, at least 64
> columns are needed for ruling out sampling being the cause for the difference
> between observations and simulations."Why?

This conclusion was poorly worded and very subjective. We have replaced this conclusion with a more precise:

*Roughly 60 days of Doppler lidar observations from the ARM-SGP site are needed to detect the differences between observed and simulated chord-length distribution with certainty.*

> Did Endo et al.  (2019) use a methodology more similar to the 1D or 3D
> method in this paper?  Do you see a change in the PDF shape from 1D to 3D
> method?  Since Endo et al.  (2019) is so recent and relevant, I suggest more
> discussions comparing your results to theirs.

When looking at the PDF below cloud base Endo et al. (2019) do not analyse clouds in terms of chords as we do. As a result, their analysis is independent of whether their PDF is sampled from 1D columns or 3D snapshots. The differences between the method adopted by Endo et al (2019) and ours are now explained in more detail in the text. We hope that these additions address the reviewer's request for clarification.

*We also note that the w PDF at cloud base studied by Endo et al. (2019) is not sampled from cloud chords but from all cloud base grid cells, so the distinction between 1D and 3D sampling has no effect on their findings.*

Although at first glance it seems as if our paper and Endo et al 2019 have a lot in common because they both analyse vertical velocities at the ARM-SGP site, they use very different approaches. We analyse everything in the context of chords with a single model setup, while Endo et al are focused on comparing different model setups and configurations. A meaningful discussion of how our work relates to theirs would require expanding our paper to include their metrics, which we would like to avoid given the already substantial length of our manuscript.

[revised manuscript text omitted]